# MRI Radiomics for Predicting Survival in Patients with Locally Advanced Hypopharyngeal Cancer Treated with Concurrent Chemoradiotherapy

**DOI:** 10.3390/cancers14246119

**Published:** 2022-12-12

**Authors:** Tiing Yee Siow, Chih-Hua Yeh, Gigin Lin, Chien-Yu Lin, Hung-Ming Wang, Chun-Ta Liao, Cheng-Hong Toh, Sheng-Chieh Chan, Ching-Po Lin, Shu-Hang Ng

**Affiliations:** 1Department of Medical Imaging and Intervention, Chang Gung Memorial Hospital at Linkou, Chang Gung University College of Medicine, Taoyuan 333423, Taiwan; 2Department of Biomedical Imaging and Radiological Sciences, National Yang Ming Chiao Tung University, Taipei 112304, Taiwan; 3Department of Radiation Oncology and Proton Therapy Center, Chang Gung Memorial Hospital at Linkou, Chang Gung University College of Medicine, Taoyuan 333423, Taiwan; 4Division of Hematology-Oncology, Department of Internal Medicine, Chang Gung Memorial Hospital at Linkou, Chang Gung University College of Medicine, Taoyuan 333423, Taiwan; 5Department of Otorhinolaryngology, Head and Neck Surgery, Chang Gung Memorial Hospital and Chang Gung University, Taoyuan 333423, Taiwan; 6Department of Nuclear Medicine, Hualien Tzu Chi Hospital, Tzu Chi University School of Medicine, Buddhist Tzu Chi Medical Foundation, Hualien 970473, Taiwan

**Keywords:** hypopharyngeal cancer, concurrent chemoradiotherapy, MRI, radiomics, overall survival, progression-free survival

## Abstract

**Simple Summary:**

MRI radiomic models outperformed traditional clinical parameters in the prediction of survival in patients with hypopharyngeal cancer who had undergone concurrent chemoradiotherapy. By combining the identified radiomic signature with independent traditional clinical variables, we were able to devise new nomograms that successfully predicted survival outcomes in this patient group.

**Abstract:**

A reliable prognostic stratification of patients with locally advanced hypopharyngeal cancer who had been treated with concurrent chemoradiotherapy (CCRT) is crucial for informing tailored management strategies. The purpose of this retrospective study was to develop robust and objective magnetic resonance imaging (MRI) radiomics-based models for predicting overall survival (OS) and progression-free survival (PFS) in this patient population. The study participants included 198 patients (median age: 52.25 years (interquartile range = 46.88–59.53 years); 95.96% men) who were randomly divided into a training cohort (*n* = 132) and a testing cohort (*n* = 66). Radiomic parameters were extracted from post-contrast T1-weighted MR images. Radiomic features for model construction were selected from the training cohort using least absolute shrinkage and selection operator–Cox regression models. Prognostic performances were assessed by calculating the integrated area under the receiver operating characteristic curve (iAUC). The ability of radiomic models to predict OS (iAUC = 0.580, 95% confidence interval (CI): 0.558–0.591) and PFS (iAUC = 0.625, 95% CI = 0.600–0.633) was validated in the testing cohort. The combination of radiomic signatures with traditional clinical parameters outperformed clinical variables alone in the prediction of survival outcomes (observed iAUC increments = 0.279 [95% CI = 0.225–0.334] and 0.293 [95% CI = 0.232–0.351] for OS and PFS, respectively). In summary, MRI radiomics has value for predicting survival outcomes in patients with hypopharyngeal cancer treated with CCRT, especially when combined with clinical prognostic variables.

## 1. Introduction

Hypopharyngeal cancer represents a distinct clinical entity, and estimates derived from the most recent update of the Taiwan Cancer Registry show that the crude incidence rate is 5.15 per 100,000 persons. Among different head and neck malignancies, hypopharyngeal cancer continues to show unfavorable survival outcomes [1]. In addition, a large proportion of patients (70–85%) have advanced stages at diagnosis due to the presence of occult symptoms and signs [2,3]. While primary surgery remains a treatment option in advanced hypopharyngeal cancer, concurrent chemoradiotherapy (CCRT) has increasingly emerged as a non-surgical alternative to achieve organ preservation [4]. Unfortunately, approximately 50% of patients with advanced hypopharyngeal cancer who had received non-surgical primary treatment ultimately experience disease recurrences [5], while the outcomes of salvage surgery are generally unsatisfactory [6,7]. In this scenario, a reliable prognostic stratification of patients treated with primary CCRT is crucial for informing tailored management strategies.

By virtue of its excellent soft tissue contrast, magnetic resonance imaging (MRI) outperforms computed tomography (CT) in terms of anatomical resolution and is commonly applied for head and neck cancer staging. Although there is a growing potential for utilizing radiomic features as prognostic biomarkers in patients with head and neck malignancies [8,9,10,11,12,13], previous research has mainly relied on CT features. Previous studies have described the robustness of CT radiomic features and their potential usefulness for predicting various clinical endpoints; however, CT radiomic features are intrinsically limited by low soft tissue contrast. It is therefore crucial to investigate the potential usefulness of MRI-based radiomics in assisting prognostic stratification. Starting from these premises, the purpose of this retrospective study was to develop robust and objective MRI radiomics-based models for predicting overall survival (OS) and progression-free survival (PFS) in patients with locally advanced hypopharyngeal cancer who had undergone CCRT. By combining the identified radiomic signature with independent clinical variables, we were able to devise new nomograms that outperformed traditional prediction models.

## 2. Materials and Methods

This retrospective study was approved by the Institutional Review Board of the Chang Gung Medical Foundation, Taiwan (reference number: 201901900B0) and received a waiver of patient consent. All procedures complied with the tenets outlined in the Declaration of Helsinki and the Good Clinical Practice Guidelines.

### 2.1. Study Patients

We retrospectively reviewed the clinical records of patients with newly diagnosed hypopharyngeal cancer who presented at the Chang Gung Memorial Hospital, Taoyuan, Taiwan, between August 2006 and September 2015. Inclusion criteria were as follows: (1) pathologically proven diagnosis of advanced hypopharyngeal cancer, (2) availability of pretreatment contrast-enhanced head and neck MRI, and (3) curative-intent treatment with primary CCRT according to the National Comprehensive Cancer Network guidelines [14]. Patients with histology types different from squamous cell carcinoma and those with second primary tumors or synchronous cancers were excluded, as were those with metastatic disease. Demographic data (including age and sex), tumor differentiation and information on clinical stages (including T stage, N stage, and overall stage) were collected in all participants. Disease staging was performed using the American Joint Committee on Cancer (AJCC) Staging Manual, Seventh Edition. A detailed description of the CCRT protocol is reported in Appendix C. For model training and validation, the study participants were divided (2:1 ratio) into a training cohort and a testing cohort.

### 2.2. Follow-Up and Survival

All patients were clinically followed-up with physical and pharyngoscopic examinations every 1–3 months during the first two years, every 3–6 months during the third year, and 6–12 months thereafter. Imaging follow-up was alternatively performed with CT or MRI every 3 months during the first post-treatment year, every 6 months during the subsequent three years, and on an annual basis thereafter. All of the imaging examinations were scheduled in advance and were generally performed in the week preceding the clinical follow-up visits. OS was defined as the interval between the date of initial pathologic diagnosis and the date of death or the day of last follow-up. Patients who were lost to follow-up or were alive at the day of last follow-up were treated as censored observations. PFS was calculated as the interval between the date of initial pathologic diagnosis and the date of the first sign of progression, death, or the day of last follow-up.

### 2.3. MRI Acquisition Protocol and Radiomic Features Extraction

MR images were acquired on a 3-Tesla scanner (Magnetom Tim Trio; Siemens Healthineers, Erlangen, Germany). After administration of a gadolinium-based contrast agent (0.1 mL/kg), post-contrast T1-weighted images were acquired using a fat-saturated turbo spin echo sequence. The following parameters were applied: repetition time/echo time = 550/10 ms, flip angle = 150°, echo train length = 3, acquisition matrix = 320 × 253, slice thickness = 4 mm, field of view = 220 mm × 220 mm, and number of averages = 2.

Using a slice-by-slice approach, all tumor volumes of interest were manually contoured on the transverse section using an open-source platform (ITK-SNAP, version 3.8.0; http://www.itksnap.org, accessed on 12 June 2019). All procedures were carried out by a senior head and neck radiologist (S.H.N.; 35 years of working experience). To assess interobserver reproducibility, images obtained from a randomly selected patient subset (*n* = 30) in the training cohort were subjected to segmentation by an independent radiologist (T.Y.S.; 8 years of working experience in the field of head and neck imaging). During tumor contouring, both radiologists were blinded to clinical information. Intraclass correlation coefficients (ICCs) were used to quantify the interobserver reproducibility of the extracted radiomic features, with reproducibility being defined as an ICC ≥ 0.75.

Prior to feature extraction, a pre-processing pipeline was applied to fat-saturated gadolinium-enhanced T1-weighted MR images to normalize signal intensity and geometric variations. The detailed procedure is described in Appendix C. An open-source platform (PyRadiomics, version 3.0.1) was used for both image pre-processing and radiomic features extraction. Most of the features extracted with PyRadiomics were in accordance with the criteria outlined in the Image Biomarker Standardization Initiative [15].

### 2.4. Model Construction and Data Analysis

Model construction and data analysis were carried out in the R environment (version 3.6.3; http://www.r-project.org/), accessed on 6 March 2020. The R packages used in the study are reported in Appendix C.

#### 2.4.1. Machine-Learning-Based Radiomic Model

Feature selection and model building were carried out in the training cohort, whereas model performance was examined in the testing cohort. We applied a feature selection strategy that included the following steps: reproducibility assessment, redundancy reduction, univariate outcome analysis, and least absolute shrinkage and selection operator (LASSO)–Cox regression modeling. We initially disregarded all features characterized by low reproducibility (i.e., ICC < 0.75), followed by removal of radiomic features showing a high degree of collinearity (i.e., Pearson’s *r* > 0.9). This was accomplished using the “caret: find Correlation” function in R. The retained features were subsequently subjected to univariate Cox analysis to preselect significant (*p* < 0.05) prognostic factors. Finally, the LASSO-Cox regression model was applied in the training set to identify the strongest predictive parameters. On the basis of the regulation weight (*λ*), LASSO shrinks all of the regression coefficients towards zero and removes irrelevant features by setting their coefficients exactly to zero. The optimal *λ* value was identified by applying a ten-fold cross-validation with minimum criteria. We finally devised a radiomic score (termed RadScore) for outcome prediction using a linear combination of the selected features weighted by their non-zero coefficients generated with LASSO. The workflow used for radiomic model construction is summarized in Figure 1.

#### 2.4.2. Development of Clinical and Combined Radiomic–Clinical Models

Clinical characteristics, including age, sex, histologic grading, T stage, N stage, and overall clinical stage, were collected from medical records and subsequently entered into a multivariate Cox regression model (with the exception of histologic grading due to a high number of missing data). Survival outcomes served as dependent variables. The combination of parameters characterized by the lowest Akaike information criterion (AIC) value was selected to construct the clinical model. The radiomic–clinical model was subsequently devised by combining the RadScore with the variables selected in the clinical model, with the resulting estimates being plotted in nomograms. Calibration curves were used to illustrate the agreement between the estimated prognosis and the observed survival.

#### 2.4.3. Statistics

The training cohort was dichotomized into two groups (high- versus low-risk) according to the median values of predicted risk scores. The same cutoffs were subsequently applied to the testing cohort. Intergroup comparisons of survival outcomes, including OS and PFS, were performed using the log-rank test. The prognostic performance of each model was assessed using iAUC based on the predicted risk. The iAUC is an integral of the product of area under the cumulative/dynamic time-dependent ROC curve and the probability density function of the time-to-event outcome [16]. Higher iAUC values reflect a better prognostic ability. The iAUC values of different models were compared, and the differences were calculated by applying a total of 1000 bootstrap replicates. All hypothesis testing was two-tailed, with statistical significance defined as a *p* value < 0.05.

## 3. Results

### 3.1. Patient Characteristics

A total of 198 patients (190 [95.6%] men; median age: 52.25 years (interquartile range = 46.88–59.53 years)) were included in the study (Figure 2). The median values of OS and PFS were 884 days (interquartile range = 331.8–2470.5 days) and 525.5 days (interquartile range = 213.8–2343.0 days), respectively. The training and testing cohorts consisted of 138 and 66 patients, respectively (Table 1). No intergroup differences were observed in terms of OS (*p* = 0.91), PFS (*p* = 0.96), age (*p* = 1.00), sex (*p* = 1.00), T stage (*p* = 0.17), N stage (*p* = 0.29), tumor grading (*p* = 0.33), and overall stage (*p* = 0.97).

### 3.2. MRI Radiomic Models

Of the 1223 radiomic features extracted from each patient, 858 were found to be reproducible (ICC ≥ 0.75) and 702 were selected for further analyses after the removal of redundancies. Univariate analysis identified 125 and 131 features as being significantly associated with OS and PFS, respectively. After applying LASSO selection, four (denoted as *f*_1_–*f*_4_) and nine (denoted as *f*’_1_–*f*’_9_) features were retained for the prediction of OS and PFS, respectively. Table 2 summarizes the selected features and their coefficients. Two RadScores—termed RadScore_OS and RadScore_PFS for the prediction of OS and PFS, respectively—were calculated through a linear combination of selected features weighted by their coefficients. The definitions of the selected radiomic features can be accessed at the following URL: https://pyradiomics.readthedocs.io/en/latest/features.html, accessed on 18 July 2021. The relationships between the RadScores and the clinicopathological characteristics in the training (Figure 3A, OS; Figure 3B, PFS) and testing (Figure 3C, OS; Figure 3D, PFS) cohorts are presented as heat maps. In the *testing* cohort, the RadScore_OS and the RadScore_PFS predicted OS (hazard ratio (HR) = 3.97; 95% confidence interval (CI) = 1.54–10.21; *p* = 0.004) and PFS (HR = 2.39; 95% CI = 1.18–4.87; *p* = 0.02), respectively (Table 3). The ability of these scores for predicting OS (model 1a: iAUC = 0.580; 95% CI = 0.558–0.591]) and PFS (model 2a: iAUC = 0.625; 95% CI = 0.600–0.633) was validated in the testing cohort (Table 4). Using the median values of the RadScore_OS (0.50) and the RadScore_PFS (−3.28) as cutoffs, the radiomic models were used to dichotomize patients in the training and testing cohorts into low- versus high-risk groups (OS in the training cohort (Figure 4A), *p* = 0.009, log-rank test; OS in the testing cohort (Figure 4B), *p* = 0.004, log-rank test; PFS in the training cohort (Figure 5A), *p* < 0.001, log-rank test; and PFS in the testing cohort (Figure 5B), *p* = 0.003; log-rank test).

### 3.3. Clinical Models

According to the least AIC values, the optimal clinical models for the prediction of both OS and PFS included T4a stage, T4b stage, and N2c stage. In the testing cohort, multivariate Cox proportional hazard analysis revealed no independent associations of T4a or T4b stages with both OS (T4a stage: HR = 1.23 [95% CI = 0.55–2.78], *p* = 0.62; T4b stage: HR = 1.33 [95% CI = 0.40–4.45], *p* = 0.64) and PFS (T4a stage: HR = 1.52 [95% CI = 0.70–3.26], *p* = 0.29; T4b stage: HR = 1.44 [95% CI = 0.48–4.32], *p* = 0.52) (Table 3). However, the associations of N2c stage with OS (HR = 2.54 [95% CI = 1.21–5.33], *p* = 0.01) and PFS were statistically significant and marginally significant (HR = 1.93 [95% CI = 0.96–3.90], *p* = 0.07), respectively (Table 3). However, these findings should be interpreted with caution due to the limited number of patients in the testing cohort. The clinical models in the testing cohort were characterized by a modest ability to predict both OS (model 1b: iAUC = 0.392 [95% CI = 0.322–0.447]) and PFS (model 2b: iAUC = 0.381 [95% CI = 0.308–0.433]; Table 4). Application of the models to the training cohort revealed a marginally significant difference in OS (*p* = 0.06, log-rank test; Figure 4C) and a statistically significant difference in PFS (*p* = 0.04, log-rank test; Figure 5C) for patients at high- versus low-risk. However, no significant difference was observed in the testing cohort (OS: *p* = 0.27, log-rank test, Figure 4D; PFS: *p* = 0.18, log-rank test, Figure 5D).

### 3.4. Combination of Radiomic and Clinical Models

In the testing cohort, multivariate Cox proportional hazard analysis revealed marginally significant associations of the RadScore with both OS (HR = 2.99 [95% CI: 0.95–9.44], *p* = 0.06) and PFS (HR = 2.09 [95% CI: 0.99–4.42], *p* = 0.05) (Table 3). The predictive performance of the combined radiomic–clinical models was successfully validated in the testing cohort for both OS (model 1c: iAUC = 0.671 [95% CI = 0.637–0.693] and PFS (model 2c: iAUC = 0.675 [95% CI = 0.641–0.687], Table 4). In both cohorts, combined radiomic–clinical models were able to stratify patients into low- versus high-risk groups (training cohort OS (Figure 4E): *p* = 0.004, log-rank test; testing cohort OS (Figure 4F): *p* = 0.04, log-rank test; training cohort PFS (Figure 5E): *p* < 0.001, log-rank test; testing cohort PFS (Figure 5F): *p* = 0.001, log-rank test).

### 3.5. Comparison of Model Performances

The differences in terms of iAUC for distinct predictive models are reported in Table 5. In general, radiomic models outperformed clinical models in the prediction of both OS (model 1a versus model 1b; iAUC difference = 0.188; 95% CI = 0.127–0.251) and PFS (model 2a versus model 2b; iAUC difference = 0.244; 95% CI = 0.181–0.3107). Compared with clinical models alone, the addition of radiomic signatures significantly improved the ability to predict both OS (model 1c versus model 1b; iAUC increment = 0.279; 95% CI = 0.225–0.334) and PFS (model 2c versus model 2b; iAUC increment = 0.293; 95% CI = 0.232–0.351). Combined radiomic–clinical models were also found to outperform radiomic models alone in the prediction of both OS (model 1c versus model 1a; iAUC increment = 0.091; 95% CI = 0.074–0.108) and PFS (model 2c versus model 2a; iAUC increment = 0.049; 95% CI = 0.038–0.059).

### 3.6. Construction of Nomograms from Radiomic–Clinical Models

With the goal of devising visual tools for predicting both OS and PFS, nomograms comprising both clinical factors and radiomic signatures were constructed (Figure 6A,B). Calibration curves (Figure 6C,D) revealed a good agreement between the predicted and observed survival endpoints (2- and 3-year OS and PFS). However, the observed outcomes showed a slight deviation from the predicted curves during the first year of follow-up.

## 4. Discussion

Using a combination of MRI radiomic signatures and clinical parameters, we were able to devise and validate prognostic models that successfully predicted OS and PFS in patients with hypopharyngeal cancer who had undergone CCRT. By applying the LASSO-Cox machine learning algorithm, a total of 13 radiomic features extracted from fat-saturated post-contrast T1-weighted MR images were found to be associated with survival outcomes. Interestingly, the integration of radiomic features improved the predictive capacity of clinical models, and the combined radiomic–clinical models showed the highest ability to predict both OS and PFS. On the one hand, our prediction tools can offer a reliable prognostic assessment suitable for clinical prognostication. On the other hand, the use of our nomograms has the potential to tailor treatment at the individual level.

Our study confirms and expands previous data on the prognostic utility of radiomic features in patients with hypopharyngeal cancer [17]. However, prior studies were conducted with heterogeneous samples in terms of disease stage, with the majority of participants being treated with surgical excision [17]. In the current investigation, we specifically focused on patients with advanced-stage hypopharyngeal cancer who had undergone CCRT. Therefore, a strength of our study lies in the possibility to obtain an accurate survival prediction in this specific subgroup.

Three first-order and seven second-order MRI radiomic features showed the highest discriminative power for prognostic purposes. It is worth noting that the prognostic features identified in our study reflected the extent of contrast enhancement observed in post-contrast T1-weighted images as being therefore related to tumor vascularity. There is ample evidence that angiogenesis has an adverse prognostic significance in several solid malignancies [18], including head and neck cancer [19,20,21]. A prior radiomic study conducted on patients with hypopharyngeal cancer who had been treated with chemoradiation demonstrated that two first-order features derived from post-contrast CT images (wavelet-LLH_firstorder_Maximum and wavelet-HLL_firstorder_Median) were independently associated with PFS [22]. Another study reported that wavelet-LHL_firstorder_Maximum and wavelet-LHL-firstorder_Kurtosis—two features extracted from post-contrast CT images—successfully predicted PFS in patients with locally advanced hypopharyngeal cancer who had undergone induction chemotherapy [23]. Finally, Li et al. [24] developed a CT radiomic signature based on first-order features (i.e., minimum, skewness, and total energy) to be used in the preoperative phase for predicting early recurrences of hypopharyngeal cancer. Second-order radiomic features—also termed texture features—reflect the statistical relationships of gray levels within an image and represent a proxy for intratumor heterogeneity. Aerts et al. [25] have previously shown that, among different CT radiomic features, those related to tumor heterogeneity had the highest value for predicting survival in lung cancer or head and neck cancer. This signature was subsequently validated in an independent cohort of oropharyngeal squamous cell carcinoma, wherein its prognostic significance was unaffected by the presence of CT artifacts [26].

Clinical decision-making in patients with malignancies is generally guided by the AJCC staging system. While the TNM stage can be considered a suitable proxy of the overall disease status, staging variables do not possess a quantitative nature and might not accurately reflect underlying differences in tumor biology. In this scenario, the use of radiomic markers has markedly potentiated our capacity to characterize highly diverse phenotypic tumor characteristics [27]. It can therefore be expected that they would possess a complementary value to traditional TNM staging for prognostic purposes. A previous study has shown that radiomic models can predict the risk of progression in hypopharyngeal cancer more effectively compared with clinical variables alone [22], an observation in line with our current data. However, a significant limitation that our investigation shares with prior studies lies in its retrospective design. This may raise questions about whether the predictive value of our combined radiomics–clinical models can still be applicable to the eighth edition of the AJCC TNM Staging Manual [28].

On examining the prognostic value of the nomograms devised in our study, we found a good agreement between predicted and observed 2- and 3-year OS and PFS; however, a slight deviation was evident when the outcomes pertaining to the first year were taken into account. Previous studies have shown that several clinical factors—different from the oncologic status—may be associated with early treatment failure in patients with locally advanced head and neck cancer who had completed their CCRT course. These variables, which include comorbidities [29,30], poor performance status [29,31], low body mass index [29,31,32], anemia [29,30], malnutrition [29,31], and low total lymphocyte count [31], are associated with impaired immune defenses and increase patient vulnerability to infectious complications during the course of treatment schemes. The lack of inclusion of these parameters in our nomograms may explain their limited ability to predict survival outcomes during the first year of follow-up.

Our study has several limitations that merit consideration. First, the reliance on manually selected slices made the extraction of MRI radiomic features labor-intensive, time-consuming, and prone to intra- and inter-observer variability [33,34]. Future studies with deep-learning-based automated segmentation techniques should work to address this limitation. Second, a certain degree of technical variability and the potential occurrence of image artifacts are still a concern in the field of radiomics. Previous studies have shown that image noise and texture may be affected by variations in MRI acquisition parameters [35,36,37,38]. Additionally, head and neck MRI is prone to swallowing-related motion artifacts. Collectively, these potential confounders may affect the prognostic value of the extracted radiomic features. Third, we solely focused on features extracted from post-contrast T1-weighted images, and other MRI sequences were not taken into account. Future research should include additional MRI sequences or multiple imaging modalities to examine a higher number of features. Fourth, it is also possible that the small sample size may have limited the power to detect significant associations and, for that reason, larger prospective cohort studies are required. Finally, the single-center design might have limited the external validity of the results. The prognostic value of our tools needs to be independently tested in larger, longitudinal investigations.

## 5. Conclusions

In conclusion, we were able to obtain an accurate prediction of survival outcomes in patients with hypopharyngeal cancer treated with CCRT through combined radiomic–clinical models and related nomograms. Our results suggest that the extraction of radiomic features from MR images may improve the prognostic stratification informed by traditional clinical variables. Integration of clinical and radiomic signatures may have the potential to tailor treatment at the individual level.

## Figures and Tables

**Figure 1 cancers-14-06119-f001:**
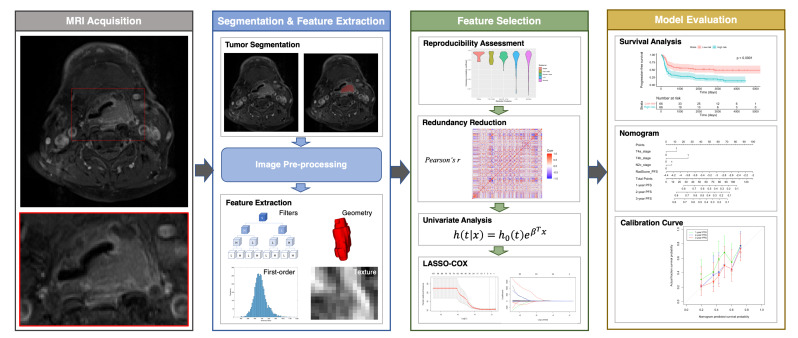
Workflow used for analysis of radiomic features. Abbreviation: LASSO—least absolute shrinkage and selection operator.

**Figure 2 cancers-14-06119-f002:**
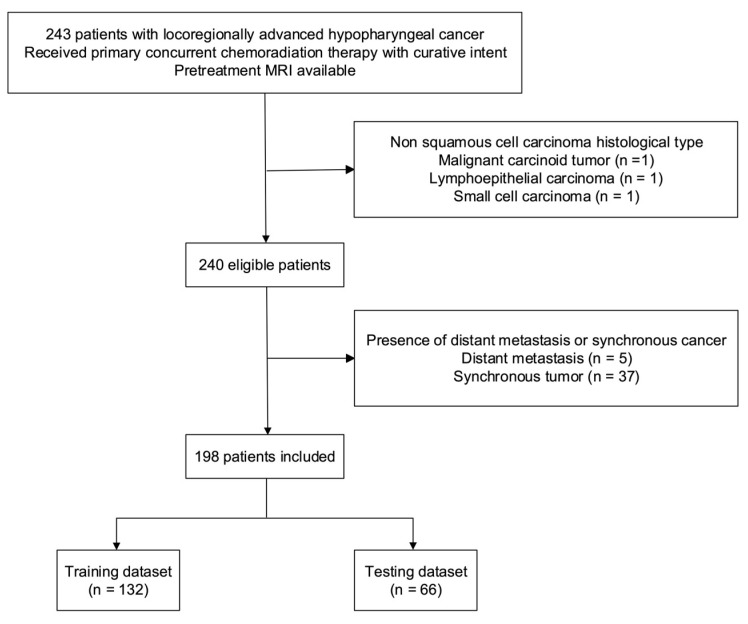
Flow of patients through the study. Abbreviation: MRI—magnetic resonance imaging.

**Figure 3 cancers-14-06119-f003:**
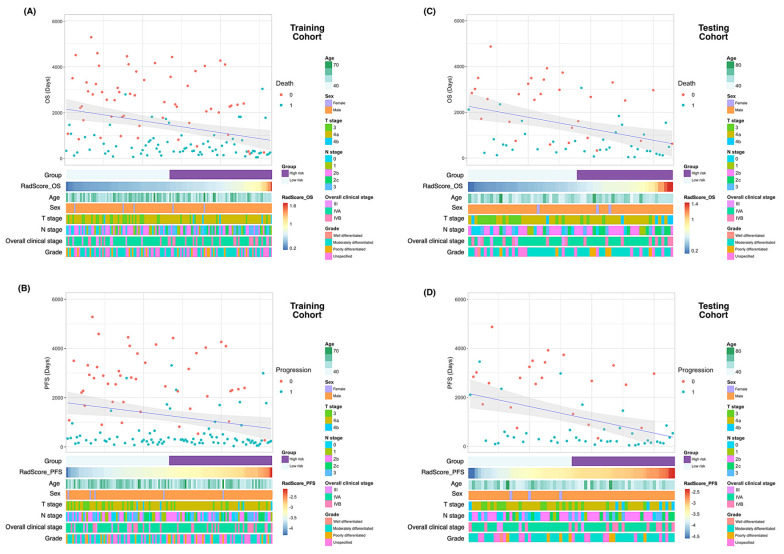
Heatmaps of clinicopathological characteristics and survival outcomes in patients stratified according to RadScores. Patients are displayed in an ascending order with respect to RadScores. Differences in overall survival and progression-free survival in the training (**A** and **B**, respectively) and testing (**C** and **D**, respectively) cohorts. Abbreviations: OS—overall survival; PFS—progression-free survival.

**Figure 4 cancers-14-06119-f004:**
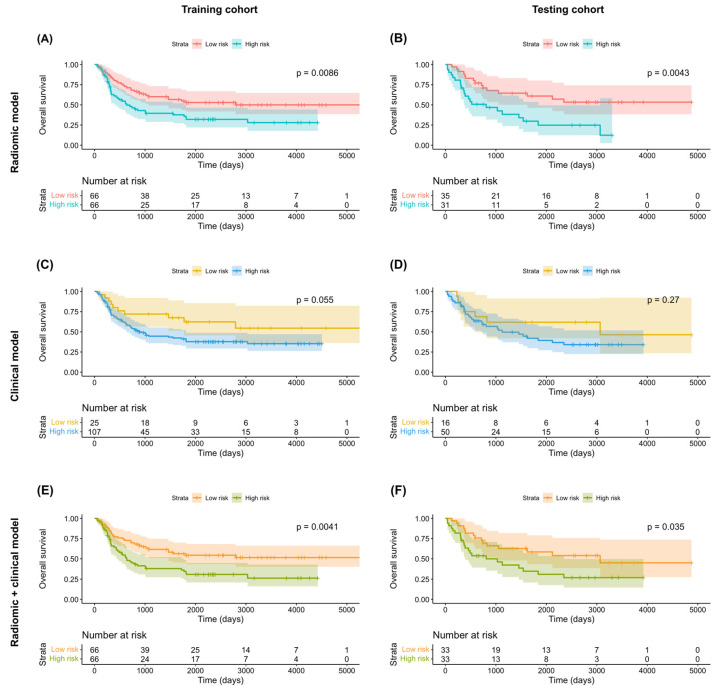
Kaplan–Meier overall survival plots in the training (**A**,**C**,**E**) and testing (**B**,**D**,**F**) cohorts stratified according to the predicted risk derived from radiomic models (**A**,**B**), clinical models (**C**,**D**), and combined radiomic–clinical models (**E**,**F**). *p* values were calculated using log-rank tests.

**Figure 5 cancers-14-06119-f005:**
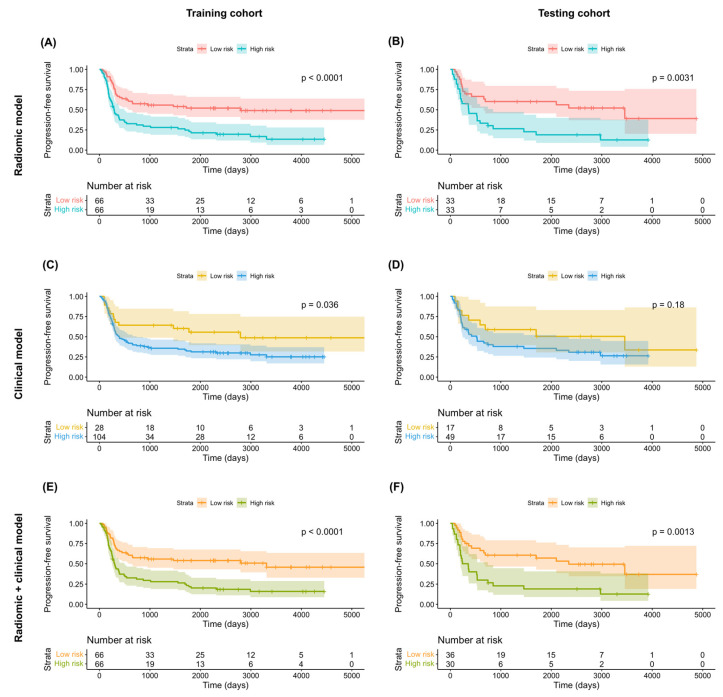
Kaplan–Meier progression-free survival plots in the training (**A**,**C**,**E**) and testing (**B**,**D**,**F**) cohorts stratified according to the predicted risk derived from radiomic models (**A**,**B**), clinical models (**C**,**D**), and combined radiomic–clinical models (**E**,**F**). *p* values were calculated using log-rank tests.

**Figure 6 cancers-14-06119-f006:**
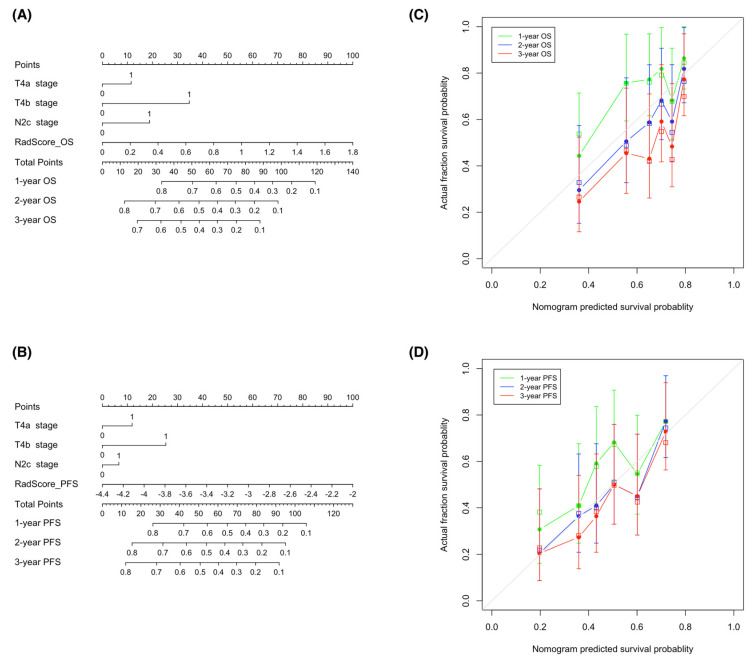
Nomograms used for the prediction of overall survival (**A**) and progression-free survival (**B**) in patients with hypopharyngeal cancer who had been treated with primary chemoradiotherapy. Calibration curves were applied to assess the predictive performance with respect to 1-, 2-, and 3-year overall survival (**C**) and progression-free survival (**D**) in the training cohort. The survival outcomes predicted by the nomogram are plotted on the *x*-axis, whereas the observed outcomes are reported on the *y*-axis. The gray lines denote ideal nomograms. The vertical bars are the 95% confidence intervals, whereas unfilled square box markers indicate bootstrap-corrected estimates. Abbreviations: OS—overall survival; PFS—progression-free survival.

**Table 1 cancers-14-06119-t001:** General characteristics of patients in the training and testing cohorts.

Variable	Training Cohort	Testing Cohort	*p* ^†^
Number of patients	132	66	
Median (IQR) overall survival, days	891 (317.3–2351.8)	851 (380.3–2599.5)	0.91 ^‡^
Number of deaths	74 (56.06)	37 (56.06)	
Median (IQR) progression-free survival, days	467.5 (216.3–2294.3)	560.5 (208.3–2553.5)	0.96 ^‡^
Number of patients with progressive disease	86 (65.15)	43 (65.15)	
Median (IQR) age, years	51.6 (46.3–60.4)	53.1 (48.5–58.4)	0.59
Sex			0.80
Male	127 (96.21)	63 (95.45)	
Female	5 (3.79)	3 (4.55)	
Clinical stage T			0.17
T3	28 (21.21)	17 (25.76)	
T4a	98 (74.24)	42 (63.64)	
T4b	6 (4.55)	7 (10.61)	
Clinical stage N			0.29
N0	24 (18.18)	10 (15.15)	
N1	18 (13.64)	4 (6.06)	
N2a	0 (0.00)	0 (0.00)	
N2b	49 (37.12)	31 (46.97)	
N2c	17 (12.88)	12 (18.18)	
N3	24 (18.18)	9 (13.64)	
Overall stage			0.98
Stage III	9 (6.82)	4 (6.06)	
Stage IVA	96 (72.73)	48 (72.73)	
Stage IVB	27 (20.45)	14 (21.21)	
Tumor differentiation			0.33
Well differentiated	2 (1.52)	1 (1.52)	
Moderately differentiated	66 (50.00)	42 (63.64)	
Poorly differentiated	18 (13.64)	7 (10.61)	
Unknown	46 (34.85)	16 (24.24)	

Data are expressed as counts and percentages in parentheses, unless otherwise indicated. Percentages may not equal 100 due to rounding. Abbreviation: IQR—interquartile range. ^†^ Student’s *t*-test for continuous variables and the χ^2^ test for categorical variables, unless otherwise indicated. ^‡^ Log-rank test.

**Table 2 cancers-14-06119-t002:** Radiomic features associated with overall survival and progression-free survival and coefficients selected by LASSO-Cox regression.

Survival Outcome and Radiomic Features	Coefficient
Overall survival	
*f*_1_: log.sigma.1.5.mm.3D_firstorder_90Percentile	8.3852 × 10^−1^
*f*_2_: log.sigma.1.mm.3D_firstorder_Energy	1.0157 × 10^4^
*f*_3_: log.sigma.1.mm.3D_firstorder_TotalEnergy	5.1365 × 10^−19^
*f*_4_: wavelet-LHL_glszm_SizeZoneNonUniformity	1.8947 × 10^−4^
Progression-free survival	
*f*’_1_: log.sigma.1.5.mm.3D_firstorder_90Percentile	4.8540 × 10^0^
*f*’_2_: log.sigma.1.5.mm.3D_glcm_SumEntropy	−2.2024 × 10^−1^
*f*’_3_: log.sigma.1.mm.3D_firstorder_Energy	5.3927 × 10^−5^
*f*’_4_: log.sigma.2.mm.3D_glcm_SumEntropy	−3.9874 × 10^−1^
*f*’_5_: log.sigma.2.mm.3D_ngtdm_Busyness	2.7371 × 10^−1^
*f*’_6_: original_glszm_SmallAreaEmphasis	−1.8426 × 10^0^
*f*’_7_: wavelet-LHH_glszm_SizeZoneNonUniformityNormalized	−7.4020 × 10^−2^
*f*’_8_: wavelet-LHL_glszm_SizeZoneNonUniformity	1.7324 ×10^−5^
*f*’_9_: wavelet-LLL_glcm_Imc1	−2.8368 × 10^0^

Abbreviations: glszm—gray-level size zone matrix; glcm—gray-level co-occurrence matrix; and ngtdm—neighboring gray tone difference matrix. LHH, LHL, and LLL denote the high- and low-pass filters on the x, y, and z dimensions, respectively (H—high; L—low).

**Table 3 cancers-14-06119-t003:** Cox Proportional Hazard Models for Overall Survival and Progression-free Survival in Patients with Hypopharyngeal Cancer Underwent Concurrent Chemoradiotherapy in the Testing Cohort.

	Overall Survival			Progression-Free Survival		
	HR	95% CI	*p* Value	HR	95% CI	*p* Value
Radiomic Model						
RadScore	3.97	1.54–10.21	0.004	2.39	1.18–4.87	0.02
Clinical Model						
T4a stage	1.23	0.55–2.78	0.62	1.52	0.70–3.26	0.29
T4b stage	1.33	0.40–4.45	0.64	1.44	0.48–4.32	0.52
N2c stage	2.54	1.21–5.33	0.01	1.93	0.96–3.90	0.07
Combined Radiomic–Clinical Model						
RadScore	2.99	0.95–9.44	0.06	2.09	0.99–4.42	0.05
T4a stage	1.06	0.46–2.44	0.88	1.36	0.62–2.96	0.44
T4b stage	0.84	0.22–3.24	0.80	1.03	0.32–3.33	0.97
N2c stage	1.92	0.84–4.37	0.12	1.69	0.59–0.82	0.15

Abbreviations: HR—hazard ratio, CI—confidence interval.

**Table 4 cancers-14-06119-t004:** Performance of different models in the prediction of overall survival and progression-free survival.

Survival Outcome	Model Type	iAUC (95% CI)
Overall survival		
Model 1a	Radiomic model	0.580 (0.558–0.591)
Model 1b	Clinical model	0.392 (0.322–0.447)
Model 1c	Combined radiomic–clinical model	0.671 (0.637–0.693)
Progression-free survival		
Model 2a	Radiomic model	0.625 (0.600–0.633)
Model 2b	Clinical model	0.381 (0.308–0.433)
Model 2c	Combined radiomic–clinical model	0.675 (0.641–0.687)

Abbreviations: iAUC—integrated area under the time-dependent receiver operating characteristic curve, CI—confidence interval.

**Table 5 cancers-14-06119-t005:** Comparison of different models in the prediction of overall survival and progression-free survival.

Survival Outcome	Model 1	Model 2	iAUC Difference (95% CI)	*p* Value
Overall survival	Radiomic model	Clinical model	0.188 (0.127–0.251)	<0.001
	Combined radiomic–clinical model	Clinical model	0.279 (0.225–0.334)	<0.001
	Combined radiomic–clinical model	Radiomic model	0.091 (0.074–0.108)	<0.001
Progression-free survival	Radiomic model	Clinical model	0.244 (0.181–0.307)	<0.001
	Combined radiomic–clinical model	Clinical model	0.293 (0.232–0.351)	<0.001
	Combined radiomic–clinical model	Radiomic model	0.049 (0.038–0.059)	<0.001

Abbreviations: iAUC—integrated area under the time-dependent receiver operating characteristic curve, CI—confidence interval. The iAUC difference was calculated as follows: iAUC (model 2)-iAUC (model 1). The 95% CIs and *p* values were calculated by applying a total of 1000 bootstrap replicates.

## Data Availability

The data presented in this study are available on request from the corresponding author.

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
