# Peer review of "MRI Radiomics for Predicting Survival in Patients with Locally Advanced Hypopharyngeal Cancer Treated with Concurrent Chemoradiotherapy"

_cancers, 2022, doi:10.3390/cancers14246119_

Round 1

Reviewer 1 Report

This paper shows that features extracted from MRI radiomics achieves higher AUC value than clinical features using Lasso-COX model.

The real main concern is that the paper is a little difficult to follow. It would be a lot easier, if domain-specific terms such as "progression-free" are described within the paper. Also, the paper used a lot of abbreviations; using a table at the beginning to summarize the abbreviations and their meaning, might make it easier to read. Also there are multiple paragraphs that provides p-values/confidence intervals, which might also be better to organize into a table.

Reviewer 2 Report

Thank you for the opportunity to review the manuscript entitled, " MRI Radiomics for Predicting Survival in Patients with Locally Advanced Hypopharyngeal Cancer Treated with Concurrent Chemoradiotherapy."  The clinical topic is important. However, I have several comments to improve the quality of the manuscript.

1. Regarding appendix A.2, how often were patients followed by physical exams? Did those happen at the same time as CT and MRI scans?

2. In the introduction, could the authors include numbers on prevalence and/or incidence for relevant country/countries?

3. Could the authors expand on previous research that has relied on CT features in the introduction? Pros, Cons, size, performance, etc.?

4. Could the authors expand on what demographic data was included already in section 2.1?

5. For figure 1, could the authors make the MRI image larger? It is difficult to see and is important to understanding where the radiomic features come from.

6. Instead of mentioning mean and range, could the authors either report mean and standard error, or median and IQR/range?

7. The authors mention that they found no inter-group differences due to non-significant p-values. However, the authors should highlight that this may be due to the low sample size and interpretation should be done cautiously. 

8. Please report IQRs together with medians in Table 1

9. The utility of reporting results of the training cohort is unclear to me. Would it be fine just to report results from the testing cohort? 

10. The paper lacks a clear and independent limitations section. Could the authors include this? That should include information on the small sample size, possible lack of generalizability due to only testing in one cohort, and possibly other things

Round 2

Reviewer 2 Report

The authors have done a nice job responding to my comments. The revised paper is much easier for me to follow. I do not have additional comments.